# A Survey of Machine and Deep Learning Methods for Privacy Protection in the Internet of Things

**DOI:** 10.3390/s23031252

**Published:** 2023-01-21

**Authors:** Eva Rodríguez, Beatriz Otero, Ramon Canal

**Affiliations:** Department of Computer Architecture, Universitat Politècnica de Catalunya, 08034 Barcelona, Spain

**Keywords:** cybersecurity, deep learning, IoT networks, machine learning, privacy

## Abstract

Recent advances in hardware and information technology have accelerated the proliferation of smart and interconnected devices facilitating the rapid development of the Internet of Things (IoT). IoT applications and services are widely adopted in environments such as smart cities, smart industry, autonomous vehicles, and eHealth. As such, IoT devices are ubiquitously connected, transferring sensitive and personal data without requiring human interaction. Consequently, it is crucial to preserve data privacy. This paper presents a comprehensive survey of recent Machine Learning (ML)- and Deep Learning (DL)-based solutions for privacy in IoT. First, we present an in depth analysis of current privacy threats and attacks. Then, for each ML architecture proposed, we present the implementations, details, and the published results. Finally, we identify the most effective solutions for the different threats and attacks.

## 1. Introduction

IoT comprises billions of connected smart devices around the world that exchange information with minimal human intervention. IoT is progressing at an enormous pace with an estimated 27 billion IoT devices by 2025 (i.e., almost four IoT devices per person) [1,2]. IoT provides real-life smart applications that improve quality of life and they have become part of our daily activities. Wearable tools and gadgets help us monitor and take care of our health, vehicles interact with traffic centers and other vehicles to improve safety, and different home elements and appliances enhance our quality of life. The significant increase in the number of IoT devices, as well as the success of IoT services and applications, have contributed to the rapid growth in the amount of generated data. The International Data Corporation report has estimated that the amount of data will increase from 4 to 140 zettabytes between 2020 and 2025 [3,4]. The significant amount of personal data collected and shared by IoT is posing increasing concerns for user privacy. In this line, several recent reports identify the different security and privacy threats associated. Gartner [5] estimates that around 15 billion smart devices will be connected to the computing network by the end of 2022. Not only could the devices be vulnerable but the enormous amount of not-secured data collected and stored online is a liability [6].

Privacy has become one of the major concerns due to the difficulty for users to control the data shared by their devices. IoT solutions must guarantee a high-level protection of user’s personal data. A data breach or data leakage in a corporation can provide confidential data to competitors causing a huge economic loss. In government systems, it can result in a sever threat for personal and medical data, or even for national security. These concerns affect IoT in different aspects such as privacy protection regulations, and processes and protocols for data management. In this context, a recent initiative lead by the IoT industry is the adoption of the EU General Data Protection Regulation (GDRP) [7] put in place in 2018. The regulation aims to give control of personal data back to users, providing fundamental directions to accomplish equitable treatments of end users and third parties. The regulation also changes the way of handling data and creates standards for the protection of user data. It addresses important privacy issues: what kind of user data can be collected and processed, under which circumstances, the purpose of collecting sensible data, the retention period, and the information provided to users. IoT is in the spotlight due to its strong dependence on user data collection and sharing. The GDPR aims to move traditional privacy solutions from focusing on service providers to users, as it promotes to improve user involvement in protecting their data through the control of what is collected, when, by who and for what purposes. In 2023, government regulations will require organizations to ensure consumer privacy rights to five billion citizens and over the 70% of GDPR [5].

Traditional privacy protection solutions have shown to be ineffective in IoT systems as emerging IoT applications require new and efficient techniques such as distributed cybersecurity controls, models, and decisions. These new solutions must take into account system development flaws, increasing attack surfaces and malicious users to detect cyberattacks. The proliferation of machine learning and hardware technology advancement could pave a way to detect the current level of sophistication of cyber-attacks in IoT networks. This leads researchers to consider ML techniques to improve the privacy of sensitive data in IoT since they provide better detection of novel cyberattacks and accuracy [8]. Moreover, ML techniques assist in the collection and processing of the vast amount of data generated in IoT to strengthen the protection of user data. ML-based solutions also can improve how sensitive data are shared between the different IoT components to keep sensitive data protected and private. Thus, they enhance the privacy protection operations and guarantee the regulatory compliance. This paper bridges the gap between ML and privacy. It provides, a comprehensive survey of ML techniques relevant to privacy protection in IoT. This survey aims to:Identify and classify privacy threats in IoT environments.Identify the challenges for successful application of ML to privacy protection.Provide a complete review of research work that applies ML techniques for privacy protection in IoT.Identify the most important and promising directions for further study.

To the best of our knowledge, previous work (reviewed in Section 2) only addresses partially the aims of this work. The rest of the survey is outlined as follows. Section 3 identifies and provides a classification for privacy threats in IoT environments. Section 4 reviews existing ML-based solutions for IoT privacy protection. Finally, Section 5 concludes the survey, and discusses challenges and future directions for privacy preservation in IoT systems.

## 2. Related Work

In recent years, privacy, ML, and IoT have crossed paths. Given the significance of ML and cybersecurity in IoT environments, several researchers have conducted surveys and tutorials (listed in Table 1) to provide a practical guide to address cybersecurity threats and a roadmap for future solutions. However, most of the existing surveys address cybersecurity in IoT environments disregarding privacy. In general, they give more importance to other cybersecurity threats as network attacks or software-based attacks. On the other hand, the few privacy focused surveys published offer a narrow view as they focus only on specific privacy preserving solutions.

### 2.1. ML-Based Cybersecurity

Several surveys analyze ML-based cybersecurity solutions but with little content on privacy. Al-Garadi et al. [9] reviewed ML-based IoT solutions. First, they provided an analysis of the vulnerabilities and attack surfaces. Then, they discussed the advantages and shortcomings of using the different ML models in each IoT layer. This work mainly focuses on ML-based security solutions. In the privacy domain, they briefly reviewed studies that demonstrate that ML algorithms can leak data [10]. This makes privacy-preserving ML and DL algorithms vulnerable to dominant attacks, for example, distributed DL methods that are unable to maintain the training set private. This survey concludes that privacy preservation is still in its infancy and needs further investigation. Hussain et al. [11] reviewed ML-based solutions for IoT cyberattacks in the three different areas: network, malware and privacy. In the privacy area, they reviewed works based on Federated Learning (FL), an Artificial Intelligence (AI) model development framework distributed over edge devices that provides secure models to maintain user privacy while improving performance. They focused on the health-care domain based on adaptive access control mechanisms, i.e., Ciphertext Policy Attribute-based Encryption (CP-ABE) and dual encryption and Merkle Hash Trees (MHTs). In the same vein, Waheed et al. [12] provided a broad overview of ML-based cybersecurity solutions. In the privacy area, they differ from the previous work since they reviewed Blockchain (BC) technologies to improve user privacy. They propose a taxonomy for security and privacy solutions in IoT using ML algorithms and BC techniques and analyze their future integration. They categorized privacy attacks in Man-in-the-middle (MiTM) and Data Privacy. ML-based solutions reviewed include Stochastic Gradient Descent (SGD), LR, Oblivious Evaluation of Multivariate Polynomial (OMPE), SVM and Naive Bayes (NB). Khan et al. [13] also reviewed ML-based cybersecurity solutions on 5G networks. They reviewed recent studies that address privacy issues related with the usage of user private data by service providers. Yet, this survey simply gives some guidelines on the usage of private data to mitigate such privacy issues. Dixit and Silakari [14] focused their review on DL models. The authors analyzed existing cybersecurity threats and DL-based solutions. In the privacy area, they reviewed adversarial attacks that raised concerns as to whether DL can violate user privacy, as well as attacks where adversaries can hack the Deep Neural Network (DNN) giving false inputs and causing misclassifications. They present existing solutions [15] based on defensive distillation and targeted gradient sign methods. Rodriguez et al. [16] surveyed DL-based solutions for all different types of cyberattacks in mobile networks. In the privacy area, they review DL-based privacy preservation works, classifying them in three main groups: collaborative DL, differential privacy, and training on encrypted data. They concluded that the vast majority of these works use supervised learning. Collaborative DL solutions only share a subset of their data, and if differential privacy is used, it avoids the disclosure of datasets. They also showed that the solutions that train DNNs on encrypted datasets achieve acceptable levels of accuracy, but poor performance (i.e., four or five orders of magnitude slower than training with non-encrypted data). Finally, Gosselin et al. [17] focused their study on privacy and security issues related to Federated Learning (FL). They reviewed unintentional data leakage, model reconstruction attacks, Generative Adversarial Network (GAN)-based inference attacks and inference attacks. Their analysis includes privacy-preserving strategies such as gradient noise addition and compression, Secure Multi-Party Computation (SMC), Differential Privacy and Blockchain technology. They conclude that inference-based attacks are the most critical to the privacy of FL. It is worth stressing that all these surveys come to the same conclusion, privacy in IoT environments is still in its infancy.

**Table 1 sensors-23-01252-t001:** Summary of existing surveys related to ML and DL for privacy protection.

Publication	Summary	Scope	
		Area	Scenario
Al-Garadi [9] (2020)	Survey of ML and DL methods for IoT Security.	Cybersecurity ML	IoT
Hussain [11] (2020)	Survey of ML and DL based security solutions for IoT networks.	Cybersecurity ML	IoT
Waheed [12] (2020)	Survey of security and privacy using ML and BC in IoT.	Cybersecurity ML	IoT
Khan [13] (2020)	Survey of the technologies used to build the 5G security and privacy model.	Cybersecurity ML	5G
Dixit [14] (2021)	Survey of DL algorithms for cybersecurity applications.	Cybersecurity DL	IoT
Rodriguez [16] (2021)	Survey of DL based cybersecurity solutions for mobile networks.	Cybersecurity DL	Mobile networking
Gosselin [17] (2022)	Privacy and Security in Federated Learning: A Survey.	Cybersecurity FL	IoT
Rigaki [18] (2020)	Survey of privacy attacks against machine learning.	Privacy attacks ML	IoT
Tanuwidjaja [19] (2019)	Survey of privacy-preserving DL techniques.	Privacy-preserving DL	Mobile networking
Boulemtafes [20] (2020)	Survey of privacy-preserving techniques for DL.	Privacy-preserving DL	Mobile networking
Liu [21] (2021)	Survey that reviews interactions between privacy and machine learning.	Privacy ML	Mobile networking
Zheng [22] (2019)	Privacy-preserving ML review for Cloud Computing and IoT.	Privacy-preserving ML	Cloud Computing
Seliem [23] (2018)	Survey of privacy threats in IoT environments.	Privacy-preserving	IoT
Amiri-Zarandi [24] (2020)	Survey of ML-based solutions to protect privacy in the IoT.	Privacy-preserving ML	IoT
Kounoudes [25] (2020)	Survey of user-centric privacy protection approaches in IoT.	User-centric privacy-preserving	IoT
Zhu [26] (2021)	Review privacy-preserving ML training solutions in IoT aggregation scenarios.	Privacy-preserving ML Training	IoT
El Ouadrhiri [27] (2022)	Differential Privacy for Deep and Federated Learning: A Survey.	Differential Privacy FL	IoT

### 2.2. Surveys on Privacy on Non-IoT Environments

ML-based privacy protection surveys for mobile networks and cloud computing scenarios have been published in recent years. Tanuwidjaja et al. [19] reviewed privacy-preserving DL works in mobile environments. They made three categories based on the use of Homomorphic Encryption, Secure Multi-Party Computation, or Differential Privacy. The classification also differentiates between classical and Hybrid Privacy-Preserving DL. Classical methods do not consider DL, while Hybrid Privacy-Preserving methods combine privacy-preserving with DL. The authors conclude that the main challenge for privacy-preserving ML techniques is the trade-off between accuracy and complexity. Activation functions based on high degree polynomial approximations have high accuracy at the expense of computational cost, while low degree polynomial approximation gives low complexity but with worse accuracy. Boulemtafes et al. [20] provided a review of privacy preservation on DL solutions. They categorized them following a multi-level taxonomy in three main groups: (1) Privacy-Preserving model training or learning, (2) Privacy-Preserving inference or analysis, (3) release a Privacy-Preserving model. Homomorphic encryption is the most used concept for Privacy-Preserving model learning and Privacy-Preserving inference, while differential privacy is the most used for releasing a Privacy-Preserving model. They evaluate privacy solutions in terms of direct and indirect data leakage protection guarantees and privacy budget consumption for differential privacy. Liu et al. [21] conducted a survey from a different perspective; they performed a classification depending on the role of ML: private ML (ML as protection target), ML enhanced privacy protection (ML as protection tool), ML-based attack (ML as attack tool). The area with more publications is private ML and the solutions proposed are mainly based on differential privacy, while ML enhanced privacy protection is gaining attention and solutions proposed categorizing personal information and predict information leakage directly from it. Another area considered is cloud computing. Zheng et al. [22] carried out a taxonomy of the existing privacy-preserving ML approaches. They classified privacy-preserving solutions into two categories: privacy-preserving training and privacy-preserving inference. Privacy-preserving training schemes are further classified based on whether the privacy-sensitive training data samples or model parameters are transmitted for training (anonymization, cryptographic methods, obfuscation, and data synthesis). Privacy-preserving inference schemes where participants transmit unlabeled data to the coordinator for inference, are classified in CryptoNets, Multi-party computation, and ObfNet.

### 2.3. Surveys on Privacy in IoT

Several works reviewed privacy issues in IoT environments but they only focused on privacy threats and attacks. Rigaki and Garcia [18] proposed a threat model and a taxonomy for the categorization of the different types of attacks based on the different actors and assets involved. They identified the assets sensitive to privacy attacks in ML: the training dataset, the model itself, its parameters, hyper-parameters, and the architecture. The actors involved are the data owners, the model owners, and the model consumers. They analyzed the privacy-related attacks which include those that aim to gain knowledge, as training data or model information. They categorized the attacks in four main groups: membership inference, reconstruction, property inference, and model extraction. From the review conducted on the different ML models, the authors conclude that the most vulnerable are the Decision Tree (DT), Linear Regression (LR), and Support Vector Machine (SVM).

Finally, the surveys that address IoT environments mainly review specific privacy-preserving solutions. Most of them focus on differential privacy but do not considering the different learning architectures: centralized, distributed, and federated. Seliem et al. [23] review major privacy threats in IoT environments and classify them in four categories: authentication and authorization, edge computing mediators, data anonymization, and data summarization. They point out the main limitations of current solutions: performance evaluation, assessment in real-life scenarios, and data access granularity. Amiri-Zarandi [24] review centralized solutions that leverage ML in IoT privacy-preservation. First, they categorize data generated in the different IoT layers, and then review ML-based privacy solutions. The works reviewed show that the ML techniques used for privacy preservation are applied to specific data categories. The authors point out the lack of standardization and interoperability to develop more global privacy-preserving solutions. In the same vein, Kounoudes and Kapitsaki [25] analyze privacy preservation solutions to identify basic characteristics that an IoT privacy framework should satisfy to guarantee user privacy while addressing GDPR requirements. From their analysis, they propose the combination of the following techniques: (1) ML techniques to provide user privacy protection, (2) policy languages to specify user privacy preferences and to express complex policies, (3) Use of negotiation techniques to provide better services to users while preserving their privacy. Zhu et al. [26] review privacy-preserving ML works differentiating between privacy-preserving prediction solutions and training solutions. In the prediction area, the solutions adopt differential privacy, secure multi-party computation, and homomorphic encryption. Training privacy-preserving solutions are classified in collaborative and aggregation scenarios. Both scenarios adopt differential privacy, homomorphic encryption, and secret-sharing solutions. Finally, El Ouadrhiri et al. [27] review current techniques to protect user privacy in FL. They classify these techniques in three main groups: first, the techniques (k-anonymity, l-diversity, and t-closeness) that protect user privacy before making a dataset available. Second, the techniques that protect user privacy during the training phase. Third, Differential Privacy based techniques, which protect users in the three stages of training a data model. They conclude that differential privacy guarantees privacy protection in FL, but it has some drawbacks of sequential composition, since the privacy degrades as the composition time increases.

The works described in this section do not completely cover the different ML techniques used for privacy preservation in IoT environments. Some of them address all the different aspects of cybersecurity in IoT with a bigger focus on network and software security than on privacy. Other works, review privacy issues in other environments. Those works that review privacy issues in IoT environments do not consider the different learning architectures (centralized, distributed and federated) and protection techniques (encryption or differential privacy). In contrast to existing surveys, this work presents a comprehensive review of ML and DL recent advances for IoT privacy preservation considering the learning architecture (centralized, distributed and federated) and the protection technique (encryption and differential privacy).

## 3. Privacy Challenges and Threats in IoT

This section analyzes privacy challenges and threats in IoT. To this end, first, we present a general overview of IoT systems to identify the privacy threats. The IoT architecture is formed by several hierarchical layers usually categorized in four main layers [28] (see Figure 1): Perception Layer, Network Layer, Middleware Layer, and Application Layer. The Perception Layer consists of devices such as actuators and sensors. It collects information from the surroundings to control and monitor the physical environment. Sensors are used to sense temperature, humidity, air quality, motion, acceleration, etc., while actuators control the activity of physical devices such as the acceleration of a car. Devices in the Perception Layer generate an an enormous amount of data that is transferred to the Network Layer for further processing and secure routing. The Network Layer processes and transmits the data through the IoT infrastructure. Sometimes a Middleware Layer is added as a bridge between the Network Layer and the Application Layer to manage vendor specific services and application needs. The Application Layer is the top layer of the IoT infrastructure. It operates on data processed from the different IoT devices. It carries out application specific functionalities.

The huge amount of exchanged sensitive data through the IoT infrastructure, the limited resource nature of IoT devices, and the data processing conducted by IoT smart applications pose serious privacy concerns. These can be classified as follows: (1) direct threats where the attacker gains access to sensible information, and (2) indirect threats, where the attacker infers or guesses the information without gaining access to the actual data. Direct information exposure can happen at the different layers of the IoT architecture. Data leakage can be caused intentionally (malicious users, malware, virus, etc.) or inadvertently (accidental disclosure of sensitive data). The Intel Security report [29] points out that 45% of data leakage is the responsibility of employees, being 51% accidental. Malicious parties exploit system backdoors to bypass authentication mechanisms to gain access to sensitive data. The recent cyberattacks suffered by NVIDIA or T-Mobile lead malicious users access sensitive information of employees or, in the case of T-Mobile, of over 50 million people [30]. Vulnerabilities of Amazon Web Services also gave malicious users access to the sensitive information of 20/20 Eye Care patients [30]. At the Network Layer, a usual threat is the transmission of confidential information without proper encryption. For example, wearable activity tracking devices transmit data over poorly encrypted Bluetooth connections [31], or Android apps send unencrypted user sensitive data to advertiser servers [32]. In smart cities, data are shared and used throughout the smart city processes involving different stakeholders which can adversely impact citizen privacy. Finally, at the Application Layer, applications and services do not clarify what happens to the data once it has been processed [33]. Moreover, user preferences and personal information have become a valued asset that can be sold to third parties, leading privacy to be an inherent trade-off. Indirect threats are caused by the accumulation and processing of user data that leads to the leakage of personal information [34]. Privacy indirect attacks [35,36] can be categorized in six main groups: membership inference, model inversion, model stealing, property inference, re-identification, and impersonation.

Membership inference attacks focus on the properties of individual samples. In this attack, given a ML model and an individual data record, an adversary determines whether or not it was used to train the model. Privacy is then violated if including an individual in the dataset is itself sensitive. For example, discovering that a personal medical record has been used in a health-related ML model leaks information about the health of that individual, which directly threatens the identification of the individual. The first membership inference attack on ML, proposed by Shokri et al. [37], considers an attacker with a black-box query access to the model and confidence scores of the input, which are used to determine the participation of given data in training. More recently, Salem et al. [38] proposed more generic attacks at a lower cost without significantly decreasing their effectiveness, while Yeom et al. [39] analyzed the case where the attacker has access to the model and the average training loss. Model inversion attack [40] enables adversaries extract the underlying participant training data by the observation of ML model predictions. Fredrikson et al. show the most prominent attacks. The first one [41] recovers genomic information of patients inverting the model of a medicine dosage prediction task, while the second one [42] extracts instances of training data from observed model predictions. They recover an image of a face similar to a given one and reconstruct them using the confidence score of the target model. Salem et al. [43] propose a model inversion attack on online learning considering the difference in the model before and after a gradient update. Model stealing attacks [44,45] recover internal training parameters or other sensitive details of ML models. Trained models are confidential [28] as they are considered intellectual property of their owners. This is especially critical in DNNs as they memorize information about their training data. Therefore, exposing the model parameters can guide to the disclosure of training data. Tramer et al. [44] design an attack that recovers parameters of a model by the observation of its predictions. The attack finds the parameters through equation solving based on input/output pairs. Property inference attack [46] deduces specific patterns of information from a target ML model. This kind of attacks finds sensitive patterns in the training data of the model. They have been conducted in Hidden Markov Models (HMM) [46], SVM [47], and Neural Networks (NN) [48]. Re-identification attacks [49] combine data from multiple collections to re-identify a record from outsourced, public or open data records. A classic well-known re-identification attack [50] makes use of a voter registration list for re-identification of a government health record from the records released by a health insurance company. Impersonation attacks [51] monitor the communication pattern of a device and try to simulate it. The adversary access to the device and alter privacy preferences [49] to finally inject fake data in the system.

## 4. ML-Based Privacy Solutions in IoT

For some time now, ML and cybersecurity have crossed paths. Traditional privacy protection solutions do not offer enough protection for the emerging IoT applications and IoT environments. Fog-IoT frameworks consist of three main layers as depicted in Figure 2. At one end, the IoT layer consists of a collection of end devices which collect and process data. They transmit data to the next level of the hierarchy, the Fog layer which undertakes data processing and storage, and connects to the cloud. The fog layer makes the middle tier of the architecture. The upper layer, the cloud, provides reliable services and support to IoT devices and Fog nodes, especially for those tasks that require abundant resources.

Given the success of ML and DL techniques in other cybersecurity areas, they are now considered for privacy. However, ML and DL techniques are a double-edged sword. One the one hand, they provide better detection capabilities. On the other hand, they are used in IoT services and applications inadequately and raise privacy concerns when using sensitive user data. This section reviews, analyzes, and compares the works that use ML solutions for improving privacy protection in IoT systems. We classify them depending on the learning architecture: centralized, distributed and federated; and the protection technique: encryption or differential privacy. For each solution, we review the attack or threat it prevents. We also analyze the architecture with a special attention to the ML method(s) used, its implementation, the data sets used for testing, and the results achieved. Figure 3 provides the summary while Table 2, Table 3 and Table 4 provide the complete analysis.

### 4.1. Centralized Encryption-Based ML Solutions

Centralized ML-based privacy protection solutions in IoT environments take advantage of encryption techniques for privacy protection. Encryption techniques are used to preserve user privacy for both ML training and ML classification.

The encryption techniques used include homomorphic encryption (HE), multi-party computation, attribute access control and lightweight cryptographic techniques. They have high computational complexity and require huge amounts of computational resources and large storage facilities. HE is generating high interest due to its excellent privacy advantages. It provides end-to-end data privacy and it allows secure third-party computations. This has led to the development of lighter HE algorithms such as Somewhat Homomorphic Encryption (SHE) and Partially Homomorphic Encryption (PHE). SHE uses small encryption keys to reduce communication costs, while PHE limits the type of operations in order to produce smaller ciphertexts, which are applied in IoT lightweight protocols. Initial works analyze network environments where data samples and ML models belong to different parties. Data owners want to perform classification but they are reluctant to expose sensitive data to the untrusted models. Model owners, in turn, are reluctant to reveal the information of the classification model as it is a valuable asset. Therefore, to protect privacy in both sides, classification protocols apply ML classifiers over encrypted data. Initial works explore the usage of generic classifiers to construct privacy-preserving classification schemes. Bost et al. [52] constructed three protocols for privacy-preserving classification of hyperplane decision, Naive Bayes, and decision trees. De Cock et al. [53] also proposed a privacy-preserving classification protocol of decision trees. Privacy-preserving protocols use homomorphic encryption and they are subsequently adopted by IoT networks and cloud computing setups. Rahulamathavan et al. [54] conducted a privacy-preserving solution in an eHealth scenario. In their setup, classification, using SVM models, is performed in a cloud server on encrypted data. They test the solution for four different datasets, three from the UCI machine learning repository (i.e., Wisconsin Breast Cancer (WBC), Pima Indians Diabetes (PID), and Iris datasets [55]) and the fourth is the facial expression dataset JAFFE [56]. They report an accuracy of 98.24%, 86.98%, 87.33%, and 89.67%, respectively. Wang et al. [57] adopted the same solution for image classification based on multi-layer extreme learning machine (ML-ELM) [58]. They proposed a framework that protects user privacy directly classifying encrypted images with DES or AES. The framework is evaluated using the MNIST dataset. It achieved an accuracy of 79.83% for AES encrypted images and 90.44% for DES encrypted images. In an eHealth environment, Zhu et al. [59] adopted the same solution as a basis for an efficient privacy-preserving medical prediagnosis framework based on a nonlinear kernel SVM. The framework, so-called eDiag, provides a privacy-preserving classification scheme with lightweight multiparty random masking and polynomial aggregation techniques. The encrypted user query is directly solved at the service provider without decryption. The diagnosis result can only be decrypted by the user. Thus, both data and the SVM classifier are protected. The solution is tested using the PID database of the UCI machine learning repository [60]. It achieves an accuracy of 94%. Jiang et al. [61] also adopted the same solution for protecting user privacy in automated medical image diagnosis in IoT environments. The cloud server performs the diagnosis through ML algorithms over encrypted data to preserve data confidentiality and patients privacy. The authors designed efficient homomorphic comparison and division schemes based on the homomorphic encryption (SHE) scheme. SHE uses small encryption keys and provides low communication costs [62]. The framework provides functionalities for lesion detection of retinal images. It is evaluated using the retinal image datasets DR1 [63], RetiDB [64], and Messidor [65]. They achieve an Area Under the Curve (AUC) between 86% and 89%. The last presented works reduce training an execution time when compared to traditional methods such as CNN or Deep Belief Network (DBN). Yet, there is still room for improvement for plain image datasets.

**Table 2 sensors-23-01252-t002:** ML-based privacy works in IoT environments—Centralized Learning.

Reference	Attack	Privacy Protection Model	ML Model	Dataset	Results	Scenario
Rahulamathavan [54] (2014)	Data leakage	Encrytion (Classification)	SVM	WBC PID IRIS JAFFE	ACC 98.24% 86.98% 87.33% 89.67%	Cloud Computing
Wang [57] (2017)	Data leakage	Encrytion (Classification)	ML-ELM	MNIST	ACC 79.83% (AES) 90.44%(DES)	Cloud Computing
Zhu [59] (2017)	Data leakage	Encrytion (Classification)	SVM	PID	ACC 94%	IoT eHealth
Jiang [61] (2019)	Data leakage	Encrytion (Training)	Homomorphic surf and fast image matching	DR1 RetiDB Messidor	AUC [86%, 89%]	IoT eHealth

### 4.2. Distributed Learning-Based Solutions

Distributed Machine Learning (DML) [66,67] has received increasing attention in recent years for privacy preservation in distributed environments and specially in IoT. In DML, the learning model generation is done at the participant devices, while the coordinating server generates a global model and shares the ML knowledge to the IoT edge devices (distributed participants). A pioneer work in this field is conducted by Shokri and Shmatikov [66]. They develop a system for collaborative DL that preserves user privacy in data training where multiple users learn NN models based on their inputs while benefiting from other user data without sharing the inputs. The DL algorithms are based on SGD because they can be parallelized and executed asynchronously. The solution is tested considering the MNIST and SVHN datasets. It achieves an accuracy of 99.1% and 93.1%, respectively for each dataset. However, this method can only protect the properties of individual samples but not the statistical properties of groups of samples. Another relevant work is conducted by Servia-Rodriguez et al. [67]. They propose a distributed learning solution on IoT devices that preserves user privacy on internet services. The solution avoids user data flows to the cloud. The model is based on a two-step process that first analyze small datasets provided by voluntary users to create a shared model. Then, the users retrain the model locally using their personal data. Finally, each user has their own personal model. The system is based on a Multilayer Perceptron (MLP) and it is trained using the WISDM Human Activity Recognition Dataset [68], NIPS and Wikipedia datasets. The solution is evaluated for shared, local and personal models. It achieves the best results (ACC = 88%) when training the model with samples of other individuals.

The works listed above assume user data privacy preservation for the simple reason that the original data never leaves the personal device. However, it has been demonstrated as erroneous, since DML models are vulnerable to privacy inference attacks (such as membership inference or model memorizing) which focus on retrieving sensitive data from trained ML models [37]. Moreover, they are also vulnerable to model inversion attacks [41] that recover original data. These privacy concerns foster the integration of protection techniques in distributed learning such as encryption and differential privacy.

#### 4.2.1. Distributed Learning and Encryption

Encryption techniques are integrated in DML to improve privacy in IoT systems. Homomorphic encryption is the most widely used encryption mechanism. User data are encrypted before being transmitted to the fog nodes (coordinators) where training and inference is performed on ciphertexts. Phong et al. [69] designed a system based on MLP and CNN models. They test it using the MNIST and SVHN datasets. They achieve an ACC of 99.1% and 93.1%, respectively. However, their solution has two main limitations. First, all participants share a common homomorphic encryption key which implies that each participant input is revealed to the other participants. Second, when a participant is revoked, the key is no longer safe. González-Serrano et al. [70] developed secure algorithms for training ML models in an IoT eHealth scenario. The solution is based on Pallier, an additive homomorphic cryptosystem. Pallier has been shown to be more efficient than algorithms such as RSA or Goldwasser-Micali in terms of encryption and decryption efficiency [71]. They use an authorization server for computation outsourcing. Shen et al. [72] developed a privacy-preserving solution also based on the Paillier cryptosystem. They proposed a privacy-preserving SVM training scheme over blockchain based on encrypted IoT data. They use blockchain techniques to build a secure data sharing platform where IoT data are encrypted using the Paillier public-key cryptosystem and then it is saved on a distributed ledger. The solution is tested using two datasets: the Breast Cancer Wisconsin Data Set (BCWD) [73] and Heart Disease Data Set (HDD) [74] available in the UCI ML repository. The authors demonstrate the effectiveness and efficiency of the solution. They use precision and recall to evaluate the ML classifiers.

Homomorphic encryption-based solutions achieve higher privacy guarantees than differential privacy-based ones at the cost of efficiency. Fully homomorphic encryption enables complex computations on ciphertexts with a high cost in terms on computation. Partially homomorphic encryption only supports a single type of operation that requires the use of trusted third-parties or the use of inaccurate models that approximate complex equations with a single type of computation. A more recent work, in the Industrial Internet of Things (IIoT), also based on blockchain technology and distributed learning is conducted by Almaiah et al. [75]. The DL model they chose is a Variational AutoEncoder (VAE) technique for privacy, which is trained in a decentralized fashion. It uses lightweight encryption and decryption based on homomorphic encryption. The framework is validated using the IoT-Botnet and ToN-IoT datasets [76]. They achieve an ACC of 95% and the latency of the framework is up to 20 ms. Osia et al. [77] also benefited from encryption techniques to preserve user privacy. They proposed a hybrid framework based on the Siamese architecture [78] for efficient privacy-preserving analytics. The solution splits the NN into the IoT device and the cloud. The IoT device performs feature extraction, and the cloud performs classification. The main innovation of this work is the feature extractor module that achieves an acceptable trade-off between accuracy, privacy, and scalability. The solution, based on the CNN model, is evaluated for two widely used classification tasks: gender classification and activity recognition. For gender classification, the datasets chosen are IMDB-Wiki [79] and LFW [80]. It achieves an ACC of 94% and 93%, respectively. Finally, Zhou et al. [81] proposed a Ciphertext Policy Attribute-Based Proxy Re-encryption (CP-ABPRE) scheme with accountability to protect user privacy in Edge Intelligence (EI) model sharing. The model is designed to enable flexible data access. The party that wants to share the model develops the access policies that determine who can access the model and, subsequently, who encrypts them. The authors combine Proxy Re-encryption (PRE) technology with CP-ABE to enable the delegation of access rights. They also introduce accountancy and verification mechanisms to track malicious behavior of edge nodes. They use edge nodes as proxy servers to reduce re-encryption computational load and assign responsibility tracking to edge nodes to constrain their behavior.

#### 4.2.2. Distributed Learning and Differential Privacy

Differential privacy is a common technique used to protect data privacy adding perturbations to the original data. Data perturbation provides privacy preservation efficient solution with a predetermined error resulting from data modification [82]. Initial data perturbation approaches include swapping, additive perturbation, condensation, randomized response, micro-aggregation, random rotation and projection, and geometric and hybrid perturbation. However, they reduce the utility of the data due to the modifications. Privacy models define the relationship between utility and the privacy granted. Initial works that laid the foundation for algorithmic learning techniques within the framework of differential privacy were conducted by Abadi [83] and Hitaj [10]. Abadi et al. [83] trained NNs with differential privacy to avoid the disclosure of dataset information. The algorithms are based on a differential privacy enhanced SGD and they are tested it using two popular image datasets: MNIST [84] and CIFAR-10 [85]. The accuracy for a differential privacy (8,10-5) is of 97% for MNIST and 73% for CIFAR-10. Hitaj et al. [10] trained DL structures locally and they only share a subset of the parameters obfuscated via differential privacy. The distributed collaborative learning system is based on CNNs. In this case, the system achieves an ACC of 97% for the MNIST [84] and AT&T [86] datasets. These initial solutions are computationally efficient. However, the resulting ML models are inaccurate in some cases since the perturbations can reduce the quality of the training data. Another related concern is that perturbations cannot completely protect data privacy. For example, some parameters are employed in differential privacy to balance data privacy and model accuracy. Initial differential privacy works have been continued and improved in the IoT ecosystem as detailed in the following paragraphs.

Wang et al. [87] improved the DML system proposed by Servia-Rodriguez et al. [67] integrating differential privacy in an IoT environment. They develop the Arden framework for DNN-based private inference using a lightweight privacy-preserving mechanism. Sensitive information is protected by means of arbitrary data nullification and random noise addition. The solution also integrates a noisy training method that injects noise into the training data to mitigate a negative impact on the performance of the cloud side. The framework is tested using the MNIST and SVHN datasets [84]. It achieves an ACC of 98.02% and 88.12%, respectively. Inference performance is improved with respect to previous solutions reducing resource consumption over 60%. A similar work was conducted by Zhang et al. [88] on distributed IoT sensing systems. They proposed the use of the obfuscate function, also based on random noise, to preserve the privacy of training data in DML. This enables customers to safely disclose the data or models to third-party providers. The obfuscate function adds random noise to existing samples or augments the dataset with new samples. The solution is tested using the CIFAR-10 dataset [85] and the logistic regression ML model. The obfuscate function slightly reduces the accuracy of the model without increasing the performance overhead. There is an inherent tradeoff between validation accuracy and noise ratio. ACC reduces by 2% when noise is added to 50% of the samples (and 5% when it is added to all the samples). Lyu et al. [89], however, considered Random Projection (RP) to perturb original data. The aim of their solution is to embed fog computing into DL to speed up computation and reduce communication costs. The fog-embedded privacy-preserving DL framework (FPPDL) preserves privacy by means of a two-level protection mechanism. First, privacy is protected using RP by perturbing original data (yet, preserving certain statistical characteristics of the original data). Then, fog nodes train fog-level models applying Differentially Private SGD. The DL model chosen is MLP with two hidden layers using ReLU activations. The solution is tested using the MNIST and SVHN datasets [84]. It achieves an ACC of 93.31% and 84.27%, respectively. Accuracy is slightly reduced compared to the centralized framework but communication and computation costs are significantly reduced. Other solutions consider Gaussian projections [90] to improve performance of privacy-preserving collaborative learning. IoT resource-constrained participants apply independent multiplicative Gaussian Random Projections (GRP) on training data vectors to obfuscate the contributed training data. The coordinator applies DL to address the increased complexity of data patterns posed by GRP. The authors compared their solution with previous approaches that consider additive noisification for obfuscation and SVMs as ML model. The performance evaluation is conducted based on two datasets: MNIST [84] and spambase [91]. The DL models considered are MLP and CNN. The results obtained show that the proposed GRP-DNN solution maintains the learning performance up to hundreds of participants in contrast to the GRP-SVM and DP-DNN with additive noisification that significantly decrease the accuracy and increase the computing overhead.

Obfuscation techniques are also considered in DML to overcome the computational overhead of encryption solutions in high volume and massive distribution of data systems (i.e., big data). Alguliyev et al. [92] proposed a method for privacy preservation of big data in IoT environments that transforms the sensitive data into non-sensitive data. To implement the method, they propose a two-stage architecture. First, data are transformed using a denoising autoencoder (AE). The sparsification parameter is added to the objective function of the AE to minimize loss in data transformation. Second, the transformed data are classified using CNN models. The method is tested using the Cleveland medical dataset extracted from the Heart Disease dataset [74], the Arrhythmia dataset [93] and the Skoda dataset [94]. It achieves an accuracy up to 97%. Du et al. [95] focused their efforts on big data privacy in Multi-Access Edge Computing (MEC) for the Heterogeneous Internet of Things (H-IoT). They proposed a ML strategy with differential privacy in MEC. They realized this strategy through two methods to maximize query accuracy and minimize privacy leakage. The first proposed object perturbation strategy consists of adding Laplacian noise to the output value. The second method adds noise to the objective function minimizing the objective disturbance. They add noise to the data in advance and then, the edge nodes process it. The proposed method is tested using four different datasets: MNIST [84], SVHN [84], CIFAR-10 [85], and STL-10 [96]. The ML models chosen use the SGD and Generative Adversarial Networks (GANs). They achieve an accuracy between 91% and 95%.

Finally, another area which makes extensive use of differential privacy is speech recognition. IoT services use neural network-based speech recognition systems as the standard human–machine interface. They collect and transmit voice information as plaintext causing the disclosure of user privacy. In some cases, speech information is used in different types of authentication which can cause significant privacy losses. In this area, Rouhani et al. [97] proposed Deepsecure to avoid data disclosure in distributed DL networks. It uses the Yao’s Garbled Circuit (GC) protocol. It decreases by two orders of magnitude the runtime compared with homomorphic-based solutions. However, Saleem et al. [98] doubted this solution due to the efficiency, reusability, and implementation issues. Later works [83] adopted differential privacy by adding perturbation in the user data. However, they considerably reduce data availability dramatically and decrease the accuracy of NNs [99]. Ma et al. [100] improved these works proposing a secret sharing-based method for linear and nonlinear operations. The DL model used is LSTM and they design interactive protocols for each gate. The designed protocols efficiently process all the data randomly splitting the audio feature data into secret shares and reduce computation and communication overhead. The solution is validated using a proprietary dataset. They achieved an accuracy above 90%. Obfuscation-based works try to overcome the limitations of DML and encryption based DML. However, they exhibit vulnerabilities to minimality attacks [101] and foreground knowledge attacks [102,103].

**Table 3 sensors-23-01252-t003:** ML-based privacy works in IoT environments—Distributed Learning.

Reference	Attack	Privacy Protection Model	ML Model	Dataset	Results	Scenario
Shokri [66] (2015)	Data leakage	Based solutions	MLP CNN	MNIST SVHN	ACC 99.1% 93.1%	IoT
Servia-Rodriguez [67] (2017)	Data leakage Model stealing	Based solutions	MLP	WISDM NIPS Wikipedia	ACC 88%	Cloud
Phong [69] (2017)	Data leakage Membership inference	Encryption	MLP CNN	MNIST SVHN	ACC 99.1% 93.1%	IoT
Osia [77] (2020)	Data leakage Model inversion	Encryption	CNN	IMDB-Wiki MotionSense	ACC 94% 93%	IoT
Zhou [81] (2021)	EI Model Stealing Impersonation	Encryption	-	-	-	IoT EI
Abadi [83] (2016)	Data leakage Model Inversion Membership Inference	Differential Privacy	SGD	MNIST CIFAR-10	ACC 97% 73%	Distributed Network
Hitaj [10] (2017)	Data leakage Membership Property Inference	Differential Privacy	GAN	MNIST AT& T	ACC 97%	Distributed Network
Wang [87] (2018)	Data leakage Membership Property Inference	Differential Privacy	DNN	MNIST SVHN	ACC 98.02% 88.12%	IoT
Zhang [88] (2018)	Data leakage Membership Property Inference	Differential Privacy	LG	CIFAR-10	ACC 95%	IoT
Lyu [89] (2019)	Data leakage Membership Inference	Differential Privacy	MLP	MNIST SVHN	ACC 93.31% 84.27%	IoT
Jiang [90] (2021)	Data leakage Re-identification	Differential Privacy	DNN CNN SVM	MNIST spambase	ACC 90% 90%	IoT
Alguliyev [92] (2019)	Data leakage Membership Inference	Differential Privacy	SAE CNN	Cleveland	ACC 97%	IoT Big data
Du [95] (2018)	Data leakage Membership Inference	Differential Privacy	SGD GAN	MNIST SVHN CIFAR-10 STL-10	ACC [91%,95%]	IoT Big data
Ma [100] (2019)	Data leakage	Differential Privacy	LSTM	Custom	ACC 90%	IoT Speech Recognition

### 4.3. Federated Learning

Federated learning (FL) focuses on learning global models based on decentralized data. This inherently increases the levels of privacy to distributed systems. FL preserves privacy bringing the learning of ML models to the IoT (client) devices. The distributed nodes use their data to locally train ML models and communicate the parameters with the other nodes. In this way, clients can generalize their local model based on the model parameters managed by the server. In FL, participants only share the gradients for aggregation. By doing so, they increase the privacy protection of localized data. Initial works in IoT networks focus on big data analytics for sensor devices in IoT eHealth environments.

Guo et al. [104] proposed a privacy-preserving framework that uses a dynamic DL mechanism to learn and infer patterns from IoT sensor data. They developed privacy-preserving learning techniques to automatically learn patterns, as similarities, and correlations of abnormalities from big sensing data. They test their framework using the CIFAR-10 dataset [85] and a CNN model. They achieve an 85% accuracy. Furthermore, in IoT eHealth environments, Nadian-Ghomsheh et al. [105] proposed a holistic hierarchical privacy-preserving architecture, based on FL plus differential privacy and secret sharing techniques. They developed a ML technique to analyze the progress of patients recognizing accurately the range of motion while preserving user privacy. To validate the proposed framework, the authors created their own dataset of images based on Pendleton [106] which includes images from patient wrist and fingers exercises. The framework is validated for the Naive Bayes Nearest Neighbor (NBNN) [107], SVM and LSTM learning models. They achieved an ACC up to 99%. Nevertheless, FL is susceptible to different attacks such as membership inference that exploits the vulnerabilities of ML models and the retrieval of private data from the coordinating servers. Consequently, it is usually coupled with additional measures such as encryption and differential privacy.

#### 4.3.1. Federated Learning and Encryption

Federated Learning adopts encryption techniques to improve user privacy. In the IoT environment, it is commonly found in IIoT and eHealth areas. DL is used to solve data-driven problems due to its strong capabilities in identification, modeling, and classification of big data. IIoT promotes FL over distributed datasets due to the lack of large datasets. Zhang et al. [108] proposed two privacy-preserving DL schemes, DeepPAR and DeepDPA. DeepPAR, based on proxy re-encryption, preserves data privacy for each participant and inherently provides dynamic update secrecy, while DeepDPA, based on the group key management, provides backward secrecy in dynamic lightweight dynamic-preserving DL. The solution is implemented using a CNN where each participant maintains a replica of the NN and its own dataset (MNIST [84] in their proposal). The authors evaluated the performance of the model in terms of computational costs concluding that the proposed schemes are lightweight and effective. Ma et al. [109] focused their efforts on improving the homomorphic encryption scheme to prevent privacy leakage in IIoT. The main innovation of their framework is that they encrypt the model updates via an aggregated public key. To test the framework, they use the UP-FALL Detection dataset [110] which consists of raw and feature sets of human activity. They achieve an ACC of 94.28%. Arachchige et al. [111] explored the encryption of local and global model parameters to improve privacy protection. They proposed a framework named PriModChain (Privacy-preserving trustworthy ML model training and sharing framework based on blockchain) that enforces privacy and trustworthiness in IIoT data. The framework blends encryption, federated ML, Ethereum blockchain, and smart contracts. The proposed framework is tested using the MNIST dataset [84]. The ML model chosen is a CNN. Moreover, the local data are not directly shared with the distributed entities. The local model parameters are encrypted using public-key encryption and the global model parameters are encrypted using a unique session key randomly generated for each federation cycle. The framework obtains an accuracy above 80%. A different approach was conducted by Fu et al. [112]. In this case, they explored encryption techniques to securely aggregate gradients. They proposed a verifiable FL (VFL) privacy-preserving framework for efficient and secure model training for big data. The framework combines Lagrange interpolation and encryption techniques to securely aggregate gradients. The participants train the model locally and encrypt their own gradients. The solution is tested using the MNIST dataset [84]. They analyzed both MLP and CNN models. It achieves an accuracy of 92.1% and 93.7%, respectively. A different approach is adopted by Lu et al. [113]. They integrate encryption techniques and FL to enable secure and intelligent data sharing in IIoT. The novelty of this solution is the privacy-preserving data sharing mechanism that incorporates FL into permissioned blockchain. They tested the solution using two datasets: Reuters [114] and newsgroups [115]. It achieves an average AUC value of 0.918.

#### 4.3.2. Federated Learning and Differential Privacy

FL, as DL, integrates differential privacy to improve privacy protection. It has been applied to different IoT systems ranging from smart cities to IIoT. Initial works promote the adoption of differential privacy instead of encryption techniques in smart cities.

Kumar et al. [116] proposed an approach similar to [72] but they used perturbation techniques rather than encryption for transforming the original data in IoT smart cities. They proposed a Trustworthy Privacy-Preserving Secured Framework (TP2SF) that consists of three main modules: a trustworthiness module, a privacy preservation module, and an intrusion detection module. The trustworthiness module uses an address-based block reputation system to ensure that IoT nodes in the smart city are not falsified. For privacy protection, they use a two-level mechanism. The first level is a data integrity checker. It uses ePoW blockchain to perform data authentication to prevent the framework from data poisoning attacks. The second level performs data transformation. It first obtains relevant features using the Pearson Correlation Coefficient (PCC) technique and, then it transforms the features to an encoded format using the Principal Component Analysis (PCA) to counteract inference attacks. Finally, the intrusion detection module deploys an optimized gradient tree boosting system. The framework is evaluated using the BoT-IoT and ToN-IoT datasets [76]. It achieves an accuracy that ranges between 95% and 99% for the different ML algorithms analyzed (DT, NB, and RF). In the same vein but based on gradient perturbation [117] instead of encryption [113], Lu et al. [117] proposed learning multi-party data without sharing sensitive information based on asynchronous FL and self-adaptive threshold gradient compression. First, participants learn from their own data locally (i.e., asynchronously train the network). Then, the participants check whether they meet the conditions for communicating with the parameter server. If so, they upload their gradients to the parameter server. The authors perform the experiments using a custom dataset where multiple problems arise due to high mobility of edge nodes. They introduce dual-weight corrections to solve the problem of unbalanced learning. Their proposal achieves an accuracy of 93%.

In geographically distributed data systems, Chamikara et al. [118] proposed a FL solution in IoT networks, the so-called DISTPAB, to prevent privacy leaks while preserving high data utility. DISTPAB employs a data perturbation mechanism that uses multidimensional transformations and randomized expansion to improve data randomness. In the distributed system, the actual data perturbation is conducted by the distributed entities (edge/fog) while the coordinating entity is in charge of the global perturbation parameter generation. The solution is tested on six datasets of the UCI Machine Learning Repository [119] (WCDS, WQDS, PBDS, LRDS, SSDS and HPDS) and they considered five different ML models (Naive Bayes, k-nearest neighbor, SVM, MLP, J48). Experiments demonstrate that the system efficiently resists attacks without loss of accuracy.

Finally, blockchain technologies are integrated in differential privacy FL. Zhao et al. [120] proposed a blockchain-based crowdsourcing FL system to improve customer learning on IoT devices to improve the predictions on future consumption behaviors. The system is based on three main technologies: FL, blockchain distributed storage, and MEC server. Customer privacy is enforced by means of DP. The workflow of the system comprises two stages. First, customers train the initial model with data collected from their home appliances. Then, the models are signed and send to the blockchain. Second, manufacturers select organizations or users as miners for calculating the averaged model. Finally, one of the miners uploads the model to the blockchain. The novelty of this solution is that the accuracy of the FL model is improved by means of a normalization technique designed by the authors that outperforms batch normalization when features are under differential privacy protection. The framework is tested using the MNIST dataset [84] and a CNN learning model. It achieves an accuracy of 90%.

**Table 4 sensors-23-01252-t004:** ML-based privacy works in IoT environments—Federated Learning.

Reference	Attack	Privacy Protection Model	ML Model	Dataset	Results	Scenario
Guo [104] (2019)	Data leakage Membership Inference	Based on decentralized data	CNN	CIFAR-10	ACC 85%	IoT Big data
Nadian-Ghomsheh [105] (2021)	Data leakage Re-identification	Based on decentralized data	NBNN SVN LSTM	Custom	ACC 99%	IoT eHealth
Zhang [108] (2021)	Data leakage Property Inference	Based on decentralized data	CNN	MNIST	-	IIoT
Ma [109] (2022)	Data leakage Property Inference	Encryption	DNN	UP-FALL	ACC 94.28%	IoT eHealth
Arachchige [111] (2021)	Data leakage Membership Property Inference	Federated Learning Encryption Blockchain	CNN	MNIST	ACC 80%	IIoT
Fu [112] (2020)	Data leakage Model Inversion	Encryption Blockchain	MLP CNN	MNIST	ACC 92.1% 93.7%	IIoT
Lu [113] (2020)	Data leakage Membership Inference	Encryption	Two-step Distance Metric Learning Scheme	Reuters Newsgroups	AUC 0.918	IIoT
Kumar [116] (2021)	Data leakage Membership Property Inference	Differential Privacy	CNN	ToN-IoT BoT-IoT	ACC [95%,99%]	IoT Smart cities
Lu [117] (2020)	Data leakage Membership Inference	Differential Privacy	DNN	Custom	ACC 93%	IoT
Chamikara [118] (2021)	Adversarial Membership Inference	Differential Privacy Blockchain	Naive Bayes KNN SVM MLP J48	WCDS WQDS PBDS LRDS SSDS HPDS	ACC [80%,90%]	IoT
Zhao [120] (2021)	Data leakage Membership Inference Re-identification	Differential Privacy Blockchain	CNN	MNIST	ACC 90%	IoT

## 5. Lessons Learned and Future Research Challenges

Privacy is one of the major concerns in IoT networks. This area of cybersecurity is in its infancy and it requires further investigation. DL-based privacy preservation works mainly follow three different approaches: collaborative DL, differential privacy, and training on encrypted data. Collaborative DL solutions use the MLP model and participants only share a subset of their data. If differential privacy is used, it avoids the disclosure of datasets. In this case, the DL models chosen are MLP, CNN, SVM, VAE, LSTM, SGD, and GAN. Finally, the proposals that train on encrypted datasets use DNNs (usually, with two hidden layers). For architectures where centralized learning prevails, the SVM model has been widely used with fairly accurate results ranging from 86.98% to 94%.

The vast majority of the studies use distributed learning. In this area, the ML models used are: MLP, CNN, SVM, and VAE. In the works reviewed, the accuracy is in the range of 93–99.1%. On the other hand, when applying differential privacy, MLP, CNN, and SVM models are still considered plus SGD, LSTM, SVHN, and GAN models are added to the list. In these works, accuracy is between 91% and 97%. Architectures based on federated learning, which use encryption as a model to preserve privacy, chose the MLP and CNN models. They achieve an accuracy between 80% and 99%. When using differential privacy instead of encryption, MLP, CNN, SVM, and DNN models are used. Accuracy is on par with the FL plus encryption mechanisms. In the context of fog computing, the majority of the works focus on reducing the computation and communication costs in IoT devices at the cost of detection accuracy. Although these approaches are evolving and improving every day, it deserves further study in order to be able to adapt to the constant changes in edge hardware computation capabilities.

Privacy protection solutions are at an initial stage. Reducing the latency and throughput of NN training on encrypted data is a big challenge. Current systems outsource DL tasks to an external service with the appropriate expertise and computational resources without comprising user data, thus making the solution computationally efficient. However, it should also consider new alternatives (e.g., quantum computing techniques) to make the solution competitive. Future directions include parallel learning and computational cost optimization. Several efforts are on the way, such as network pruning and the interplay between different malicious activities. Finally, the evaluation and assessment of privacy solutions in real-life scenarios is still a challenge together with the trade-off between real IoT application quality-of-service and privacy protection.

## 6. Conclusions

This survey reviews and examines ML-based solutions for privacy threats in IoT systems. First, privacy threats are analyzed and classified. In direct privacy threats, attackers gain access to sensible user data. In indirect threats, the attackers infer the information without gaining access to the sensible data. Direct data leakage can happen at the different layers of the IoT architecture and it can be caused intentionally or inadvertently. Indirect attacks, which usually occur at the application layer, are classified in six main groups: membership inference, model inversion, model stealing, property inference, re-identification, and impersonation. In this survey, we analyzed ML-based privacy preservation works addressing IoT privacy attacks. This work provides a knowledge base for future technologists and researchers working in this area. For all the works, we analyze the architecture, the ML method(s) used, its implementation, the data set, and the reported results. Whenever possible, we compared the performance of the different proposals. Works addressing direct privacy data leakage attacks use encryption techniques on their datasets before training or classification. They achieve acceptable levels of accuracy. However, they are still too slow. They are about four or five orders of magnitude slower than training or classifying non-encrypted data.

Distributed and federated learning is used for preventing indirect privacy attacks. In this area, a combination of collaborative learning, encryption and differential privacy is the most used solution. From all the works analyzed, we can conclude that privacy protection solutions are at an initial stage. Finding better tradeoffs between privacy guarantees and computational costs is still a demanding challenge. From reducing outsourcing to external services to parallel processing, from more efficient ML models to low-complexity algorithms, privacy protection research requires the effort of researchers in all computing fields in the next years.

## Figures and Tables

**Figure 1 sensors-23-01252-f001:**
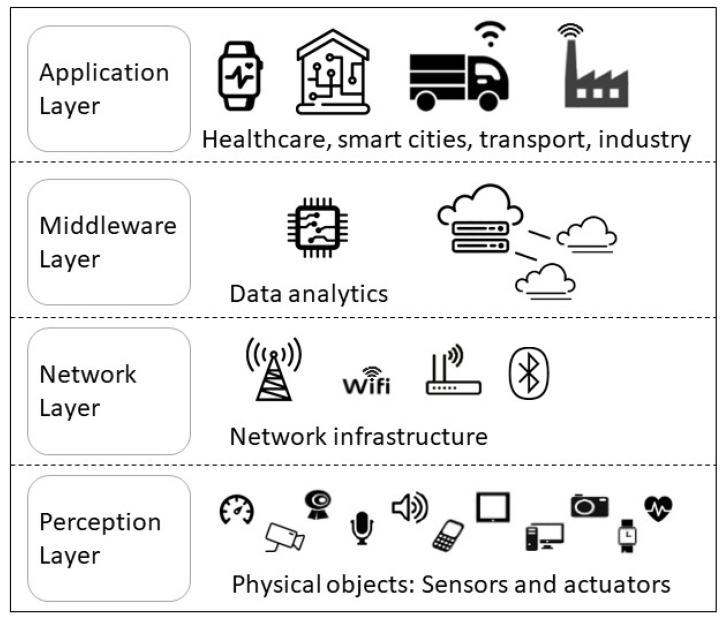
IoT architecture.

**Figure 2 sensors-23-01252-f002:**
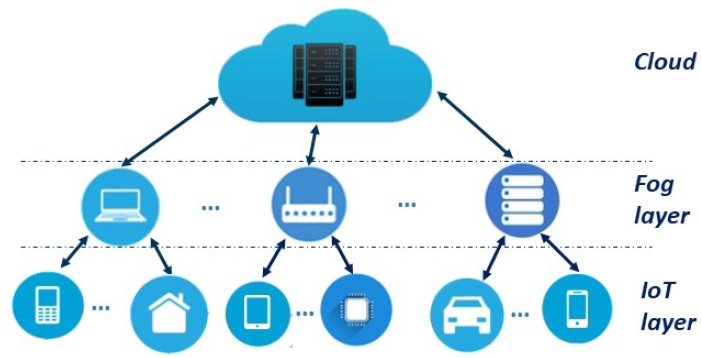
IoT system architecture.

**Figure 3 sensors-23-01252-f003:**
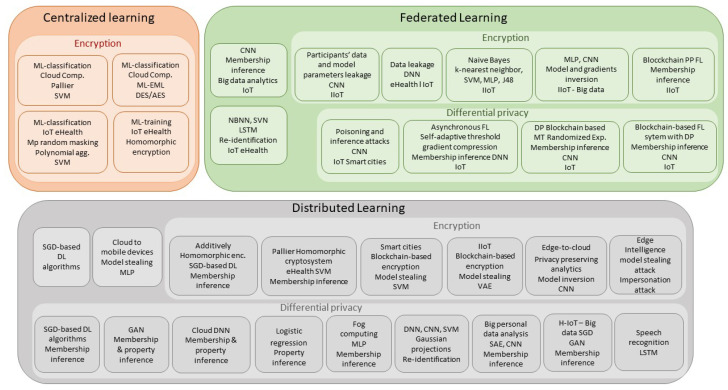
Overview of ML privacy based solutions in IoT.

## Data Availability

Not applicable.

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
