# Peer review of "A Survey of Machine and Deep Learning Methods for Privacy Protection in the Internet of Things"

_sensors, 2023, doi:10.3390/s23031252_

Round 1

Reviewer 1 Report

The authors should justify better why ML and DL are important for the privacy of the IoT. In the introduction they claim that existing privacy solutions cannot be applied in the IoT due to performance issues, however ML and DL are  not  "lightweight". 

In the related work section, the paper nicely presents existing related surveys. However, only in the last paragraph they mention a few high level differences between their work and the cited related papers. The authors should made clearer how differentiate from existing surveys. 

My biggest concern is that the majority of the cited papers are not related to "Privacy protection in the IoT"; many of them are not even related to the IoT. The authors must decide whether this paper is about IoT or Privacy and ML in general. Moreover, all papers are merely cited and a brief description of their content is provided: instead, the authors should discuss these papers more deeply and discuss better their applicability in the context of the IoT.  May be the authors should follow an approach similar to  https://ieeexplore.ieee.org/document/9072101 (note: the reviewer is not affiliated with that paper), which presents various ML and DL methods and discuss their suitability in the IoT. 

Overall, the authors should put some significant effort to re-organize their paper and discuss identified solution in the context of the "privacy of the IoT"

Minor comments

- In the introduction the section related to GDPR is not relevant
- Are you sure that in section 3, reference [20] is used correctly?
- The network layer does not necessarily transmit data securely
- Homomorphic encryption cannot be used in IoT devices

Reviewer 2 Report

The authors presented a survey on the implementation of ML and DL algorithms in IoT environments for security and issues. The main contribution claimed in the paper are

Identify the challenges to the successful application of ML to privacy protection. And

 Identify the most important and promising directions for further study

This research did not present a sufficient analysis of these points. Section 5 needs to be divided into two parts and a complete discussion on future directions should be presented.

A critical analysis on the previous research studies should be presented in sections 3 and 4. Presenting a table and summarizing the previous research studies are not sufficient as referenced papers in bibliography have very comprehensive tables and analysis.   

Different sections need English revisions as there are many grammatical mistakes.

Round 2

Reviewer 1 Report

The paper has been improved. I still have the concern that many surveyed papers are not IoT, e.g., medical images are not IoT, neither are mobile networks. May be you should try describe a reference IoT architecture. 

In your response you are writing " Recent advancements in the performance of homomorphic encryption (HE) make it possible to help to protect sensitive data in IoT systems" however in your paper you are writing "[...] They have high computational complexity and require huge amounts of computational resources and large storage facilities". How are these two combined? You should provide a reference IoT system and explain where encryption and processing of data takes place. 

Table 2 does make much sense. It is hard to read, since it is not obvious where are the boundaries of each column. In general however, I think that table is not required, the information presented there is better presented in the next tables. 

Minor

- What do you mean by "hacking skills". In general try to avoid the term "hacker". I think in most cases you want to say "malicious users"

- Change the title of sections 2.2 and 2.3. Suggested tiles: 2.2 "Surveys on privacy on non-IoT environments", 2.3 "Surveys on Privacy in IoT"

- Change the caption of Table 1, since it does not concern only IoT

Round 3

Reviewer 1 Report

For section 2.2 title it is better to use "Surveys on Privacy in non-IoT environments" ( I know that you used my previous suggestion but I think this one is more correct. Sorry for the confusion)

Although the paper can be accepted as it is, it can be enhanced if for each subsection in section 4, you provide a high level example of how each ML mechanism works, based on the reference architecture you introduced at the beginning of section 4. In any case, it is fine even if you don't do that. 

Also, please check with the editor, I believe that the emails of all authors should appear in the first page. 

I believe that the URL of reference 115 is wrong